# Impact of Dietary Lipids on the Reverse Cholesterol Transport: What We Learned from Animal Studies

**DOI:** 10.3390/nu13082643

**Published:** 2021-07-30

**Authors:** Bianca Papotti, Joan Carles Escolà-Gil, Josep Julve, Francesco Potì, Ilaria Zanotti

**Affiliations:** 1Dipartimento di Scienze degli Alimenti e del Farmaco, Università di Parma, Parco Area delle Scienze 27/A, 43124 Parma, Italy; bianca.papotti@unipr.it; 2Institut de Recerca de l’Hospital de la Santa Creu i Sant Pau & Institut d’Investigació Biomèdica (IIB) Sant Pau, 08041 Barcelona, Spain; jescola@santpau.cat (J.C.E.-G.); jjulve@santpau.cat (J.J.); 3CIBER de Diabetes y Enfermedades Metabólicas Asociadas (CIBERDEM), 28029 Madrid, Spain; 4Unità di Neuroscienze, Dipartimento di Medicina e Chirurgia, Università di Parma, Via Volturno 39/F, 43125 Parma, Italy; francesco.poti@unipr.it

**Keywords:** reverse cholesterol transport, HDL, fatty acids, sterols, rodents

## Abstract

Reverse cholesterol transport (RCT) is a physiological mechanism protecting cells from an excessive accumulation of cholesterol. When this process begins in vascular macrophages, it acquires antiatherogenic properties, as has been widely demonstrated in animal models. Dietary lipids, despite representing a fundamental source of energy and exerting multiple biological functions, may induce detrimental effects on cardiovascular health. In the present review we summarize the current knowledge on the mechanisms of action of the most relevant classes of dietary lipids, such as fatty acids, sterols and liposoluble vitamins, with effects on different steps of RCT. We also provide a critical analysis of data obtained from experimental models which can serve as a valuable tool to clarify the effects of dietary lipids on cardiovascular disease.

## 1. Implications of the Reverse Cholesterol Transport in Cardiovascular Disease

Atherosclerotic cardiovascular disease (CVD) is the leading cause of mortality worldwide. Currently, almost 18 million people die annually from CVD, representing 30% of all global deaths [1,2]. Atherosclerosis is a common mechanism of various manifestations of CVD, including coronary heart disease, heart failure, stroke, and hypertension. The retention of cholesterol-engorged macrophages in the arterial wall drives the formation of fatty lesions that can develop into mature atherosclerotic plaques, potentially leading to clinical cardiovascular events [3]. By removing excess cholesterol from extrahepatic tissues for their final excretion into the feces, reverse cholesterol transport (RCT) represents a physiological, protective mechanism. This process is mainly driven by high-density lipoproteins (HDL) and it is widely recognized as one of the main atheroprotective functions of these particles, especially when macrophage-derived cholesterol is involved [4].

The whole process consists of four main steps: (1) cholesterol efflux from macrophages to plasma HDL, (2) HDL remodeling, (3) cholesterol hepatic uptake, and (4) fecal excretion [4]. Cholesterol efflux is often considered the first and rate-limiting step of the entire process. Its efficiency relies on both the capacity of cells to release excess cholesterol and the ability of extracellular particles (mostly HDL) to accept it. The expression of lipid transporters ATP-binding cassette (ABC) A1 and G1, and scavenger-receptor class B type I (SR-BI) on the macrophage plasma membrane, and properties of HDL (i.e., their size and composition) are critical determinants of this first step. For deeper insights into these aspects, a paper by Rosenson and colleagues is recommended [5]. Once in the circulation, HDL undergo extensive transformation, driven by systemic transfer proteins that promote cholesterol esterification (lecithin:cholesterol acyltransferase (LCAT)) [6] or the exchange of lipids with apolipoprotein B (apoB)-containing lipoproteins (phospholipid transfer protein (PLTP) and cholesteryl ester transfer protein (CETP)) [7], as well as by lipases (lipoprotein lipase (LPL) [8], endothelial lipase (EL), and hepatic lipase (HL)) [9]. Such enzyme-mediated modifications will promote HDL remodeling which will become crucial for RCT efficiency, as it influences both the acceptor and donor properties of this lipoprotein class.

Cholesterol uptake by the liver may occur upon the interaction of HDL with its receptor, SR-BI [10], or through the binding of apoB-containing lipoproteins with the low-density lipoprotein receptor (LDLR) [11]. The former is the principal pathway in mice, which do not express CETP, whereas the latter is the main route for cholesterol hepatic uptake in CETP-expressing species, including humans.

Following uptake, hepatic cholesterol can be either directly transported to the intestine by the canalicular transporters ABCG5 and ABCG8 [12] or undergo a hepatic conversion into bile acids before elimination into the feces.

Beyond RCT, a new pathway involving the direct elimination of cholesterol in the intestine has emerged. This process, called transintestinal cholesterol excretion, requires lipoprotein-derived cholesterol to be taken up from the basolateral side of the intestinal cells, transported to the apical membrane and from there released into the lumen by the ABCG5/G8 transporters [13].

From an experimental point of view, numerous methods are available for the quantification of RCT in vitro, either in animal models or in humans [14,15]. Indeed, the evaluation of the in vitro cholesterol efflux capacity of HDL collected from human subjects, measuring the extent of the first step of RCT, has been proposed as a valuable biomarker of cardiovascular risk [16,17,18,19]. Conversely, other studies revealed no association between HDL efflux capacity and cardiovascular outcomes [20,21,22,23], challenging the application of this method in the clinical setting.

The whole-body RCT capacity can be assessed in humans by tracing cholesterol from the periphery to the feces. A radioisotope-based assay was recently developed by Cuchel’s group, with the advantage of following macrophage-derived cholesterol. However, the use of radioactive beta-emitters may raise safety and ethical concerns [24]. Alternatively, the use of stable isotopes has been proposed, although the evaluation of systemic, and not specifically macrophage-derived cholesterol, reduces its implications in terms of atheroprotection [25].

Certainly, the strongest evidence of the atheroprotective effect of HDL in RCT has come from studies in animal models. In particular, the assay originally developed by Rader’s group, based on the evaluation of macrophage-specific RCT (m-RCT) upon an intraperitoneal injection of radiolabeled cholesterol-loaded macrophages in mice, very closely mimics the in vivo setting and specifically tracks the most atherogenic pool of cholesterol in the body [26]. The application of this method demonstrated that m-RCT in mice inversely correlates with atherosclerosis progression [27], providing for the first time the concept that this major HDL function, more than HDL plasma levels, is a milestone for atheroprotection.

The influence of either nutritional or nutraceutical interventions on m-RCT has been widely investigated in experimental animals, in an attempt to unravel mechanisms linking the diet to cardiovascular health and to test non-pharmacological approaches for preventing cardiovascular disease [28,29,30,31].

In this review we will critically analyze studies in which the impact of dietary lipids on RCT in animal models was evaluated.

## 2. General Aspects of Dietary Lipids: Chemistry, Sources, Intake, and Effect on CVD in Humans

Dietary lipids account for about 25–45% of the total energy intake in industrialized countries and, in addition to their contribution in defining the organoleptic properties of foodstuffs, such as palatability and consistency, they exhibit key nutritional functions. Dietary lipids, indeed, by providing about 9 kcal/g, represent the most important energetic nutritional constituent [32]. Although dietary triglycerides (TGs) account for about 95% of the total lipid intake in terms of composition, free fatty acids (FFAs), which are usually found in their esterified form, as TGs, and phospholipids, account for only about 4–5%, despite their great contribution to many physiological functions [32]. Cholesterol is mostly present in its non-esterified form and accounts for about 3% of the total lipid intake. Table 1 and Figure 1 summarize the main lipid dietary sources, the average daily intake, and the content of the most representative molecules in the mentioned foodstuff, whereas the main evidence of their impact on in vivo m-RCT are represented in Figure 2 and will be discussed in the following sections.

### 2.1. Fatty Acids

Dietary lipids contain a large amount of fatty acids: from a chemical point of view, they are carboxylic acids composed of a generally linear aliphatic chain, with 4–36 carbon residues, a carboxylic terminus on one end, and a methyl group on the other [56].

Based on their chain length, saturated fatty acids (SFA) can be in turn divided into short-, medium-, long-, and very long-chain FA.

Short-chain saturated fatty acids (SCFA) present an aliphatic chain with two to six carbons [57] and they are found in a very low proportion in the diet, mainly in butter (Table 1). The largest amount of these molecules originates during saccharolytic fermentation by gut microbiota in the proximal colon of undigested or partially digested fibers and resistant starch, such as starch contained in coarse grain, raw potato flour, retrograded starch contained in potatoes, and chemically modified starch in processed food [58,59]. Other sources of SCFA are oligosaccharides such as fructooligosaccharides, mannanoligosaccharides, galactooligosaccharides, and chitooligosacchrides [60]. Accumulating studies point to a protective effect of SCFA in different CVD-related conditions by affecting glucose and lipid homeostasis and immune cell activation, as demonstrated by in vitro and in vivo evidence [61,62,63,64,65]. Furthermore, a possible regulation of systolic and diastolic blood pressure was reported in a recent meta-analysis of eighteen clinical trials [66]. Mechanistically, SCFA bind to different G-protein coupled receptors (GPCRs), such as G-protein coupled receptor 41/free fatty acid receptor 3 (GPR41/FFAR3), GPR43/FFAR2, GPR109A, and olfactory receptor 78, mainly expressed in immune cells, adipocytes, and epithelial cells [67].

Medium-chain saturated fatty acids (MCFA) present an aliphatic chain with 6–12 carbon atoms. Virgin coconut oil represents the main dietary source of MCFA (Table 1), as about 50% of the FA content is represented by lauric acid [68,69]. Moreover, MCFA can also be found as medium- and long-chain TG in human milk fat, representing a fast energy source as they are rapidly absorbed, oxidized, and digested. For this reason, milk formulas for preterm infants are usually enriched in MCFA for up to 50% of the total fat content [70]. Indeed, ingested MCFA, mainly caprylic acid and capric acid, have limited potential for storage as TGs: they are poorly incorporated into chylomicrons and rapidly enter hepatic mitochondria, where they are immediately oxidized to acetyl-coenzyme A, and subsequently converted into ketone bodies [71]. Concerning lipoprotein metabolism, a meta-analysis of different dietary compositions demonstrated that the replacement of 1% of dietary carbohydrates with lauric acid leads to a significant increase in apolipoprotein A-I (apoA-I) of 13.8 mg/L, despite a parallel, non-significant increase in apoB levels of 5.6 mg/L, leading to a reduced overall total cholesterol (TC)/HDL-C ratio [72]. However, long-term clinical trials are still needed to finally assess the impact of MCFA on cardiovascular health [73].

Long-chain saturated fatty acids (LCSFA) have an aliphatic chain with 14–18 carbon atoms and are present in both plant- and animal-derived foodstuffs (Table 1) [34]. Lauric, palmitic, and particularly myristic acid are known to exert a prominent hypercholesterolemic effect in humans, with an overall increase in TC, LDL-C, and HDL-C levels [72,74,75], whereas stearic acid seems to be neutral in respect to these parameters [76]. One hypothesis is that once absorbed, stearic acid is rapidly converted into oleic acid, whereas palmitic acid must be first elongated to stearic acid and only then desaturated into oleic acid, thus exerting their LDL-raising action for less time [76]. Finally, a prothrombotic function has been attributed to stearic acid, as a slight increase in fasting plasma fibrinogen was observed in healthy subjects, but to date, no conclusive observations are available [77]. Consistently, a recent prospective cohort study and meta-analysis involving more than 1 million participants showed an association between SFA intake, mainly of LCSFA, and coronary heart disease mortality [78].

Monounsaturated fatty acids (MUFA) present an aliphatic chain with only one C=C double bond; in particular, in MUFA in cis configuration, the two hydrogens adjacent to the double bond are on the same side of the aliphatic chain [79]. The average worldwide intake of MUFA ranges from 10% of daily total energy consumption to 22%: generally, southern European Countries had a higher MUFA intake compared to central and northern ones [80]. Other differences are related to the dietary sources: although vegetable oils, mainly olive oil, represent the main MUFA sources in Greece, Italy, and Spain by providing up to 64% of total MUFA intake [36], in other countries, MUFA are introduced through meat, meat products, added fats, and dairy products [81,82]. Oleic acid is the most important MUFA, accounting for about 92% of all MUFA introduced through the diet [37,38,39]. In recent years, various studies have analyzed the effect of olive oil on CVD-related outcomes in humans, highlighting an overall antiatherogenic and antithrombotic effect, by increasing the HDL-C/LDL-C ratio, decreasing TC plasma levels [80,83], reducing blood pressure [84], and exerting a beneficial anti-inflammatory effect [85]. However, other studies have reported a neutral or negative association between CVD and MUFA. Recently, both a Mendelian randomization analysis and a prospective cohort study reported the lack of association between CVD risk factors, CHD mortality, and serum MUFA levels [78,86]. Hence, further studies are required to clarify the links between MUFA properties and CVD.

Other MUFA are also absorbed with the diet, although in smaller amounts, and their effects on CVD have been rarely addressed. Among these, a long-chain MUFA (LCMUFA), erucic acid, mainly found in rapeseed oil, has generated many concerns due to its adverse cardiotoxic effects in animal models [87]. These could be, at least partly, explained by the predominantly peroxisomal oxidation of LCMUFA, thereby leading to the generation of lipoperoxides that attenuate mitochondrial fatty acid oxidation and glycolysis and promote apoptosis [87]. It should be noted that canola oil (Canadian oil low in erucic acid) was developed, containing <2% erucic acid, low levels of SFA (<7%), high amounts of MUFA and PUFA, plant sterols, and tocopherols [88], becoming the 3rd largest vegetable oil by volume of production worldwide after palm and soybean oil.

Polyunsaturated fatty acids (PUFA) contain two or more C=C double bonds in the cis configuration. Chemically, dietary PUFA can be divided into ω-3 and ω-6 species, based on the position of the last double bond in the methyl-terminus of the molecule [89]. In contrast to ω-3 and ω-6 PUFA, which are considered essential PUFA, ω-9 PUFA are mainly synthesized endogenously [90] in situations of linoleic acid (LA) and α-linolenic acid (ALA) deficiency.

The most important ω-3 PUFA are ALA (18:3 ω-3), stearidonic acid (18:4 ω-3), eicosapentaenoic acid (EPA; 20:5 ω-3), docosapentaenoic acid (DPA; 22:5 ω-3), and docosahexaenoic acid (DHA; 22:6 ω-3). Importantly, EPA, DPA, and DHA can be endogenously synthesized using ALA as a precursor, with EPA and DHA being the main metabolic end products [91]. However, the production rate of both EPA and DHA is rather limited in humans. In contrast, the conversion rate of EPA and DHA from ALA is much more elevated in some cold-water fatty fish; in particular, fish oils can be found as ω-3 PUFA supplements or in the form of ethyl esters or acylglycerides in a concentrated form (Table 1) [92]. Accumulated evidence suggests that increasing ω-3 PUFA-rich food consumption, through the administration of supplements or supplemented food, may confer protection against cardiovascular risks [93,94,95]. Indeed, different studies have been conducted, revealing that increased ω-3 PUFA consumption exerts a favorable influence on different CVD-related outcomes, such as myocardial infarction, stroke, arrhythmias, atherosclerosis, thrombosis, CHD, and peripheral artery disease (PAD) [92,96]. Notably, such cardioprotective effects have been frequently accompanied by either marginal changes in serum HDL-C [97,98] or reduced concentrations of HDL-C cholesterol in different human studies [96], although they have been linked to substrate competition for cyclooxygenase enzymes between ω-3 PUFA and arachidonic acids [99]. Additionally, ω-3 PUFA might counteract atherosclerotic plaque progression through their anti-inflammatory activity and by reducing the expression of adhesion molecules and platelet-derived growth factor (PDGF) [100]. Moreover, a recent meta-analysis highlighted the role of ω-3 PUFA in antioxidant defense against reactive oxygen species, possibly improving the pathological status of different diseases, including atherosclerosis [101]. Concerning the lipid profile, in an adapted dietetic regimen in which saturated fats were partially substituted with ω-3 PUFA, fasting plasma concentrations of TC and LDL-C were strongly reduced [41]. Consistently, a recent, extensive meta-analysis designed to evaluate the effects of increased ω-3 PUFA intake and different outcomes linked to CVD, such as cardiovascular events, adiposity, and the lipid profile, suggested that the increase in ω-3 PUFA may slightly reduce CHD mortality risk, arrhythmia, and CV events, together with a reduction of circulating TGs [102].

Dietary ω-6 PUFA include LA (18:2 ω-6), with its main metabolic end products being γ-linolenic acid (GLA; 18:3 ω-6), dihomo-γ-linolenic acid (DHGLA; 20:3 ω-6), and arachidonic acid (AA; 20:4 ω-6) PUFA [45]. The relevance of ω-6 PUFA, especially of LA, is also emphasized by the fact that breast milk contains a high amount of this fatty acid [103], leading to its fortification in milk formulas, to promote the correct development of newborn babies [103]. Dietary ω-6 PUFA are mainly involved in the modulation of various physiological processes, but evidence related to their potential cardioprotection is still not conclusive. Indeed, although different epidemiological studies show an inverse association between ω-6 PUFA intake and CVD risk, LDL-C, and blood pressure levels [104,105,106,107], some concerns have arisen from the observation that ω-6 PUFA manipulation may also exert a proinflammatory effect by increasing prostaglandin and leukotriene production [108]. The major effect attributed to ω-6 PUFA, especially to LA, is the reduction of TC, especially the LDL-C fraction, in healthy or moderately hypercholesterolemic subjects [109,110,111,112], with the latter possibly being caused by the activation of peroxisome proliferator-activating receptor (PPAR) γ and α, which upregulate the hepatic expression of LDL receptor, promote the reduction of proprotein convertase subtilisin/kexin type 9 (PCSK9) levels and downregulate apoB synthesis [113,114,115,116]. Moreover, the available data from both intervention and epidemiological studies highlight a possible association between dietary ω-6 PUFA intake and lower circulating inflammatory markers [117,118], although the exact molecular mechanisms are still under investigation [108].

Trans fatty acids (TFA) are unsaturated fatty acids with one or more C=C double bonds in the trans geometric configuration; plasma and tissue levels of TFA completely reflect dietary intake, as humans are unable to synthesize them [119]. The involvement of TFA in CVD development has been comprehensively reviewed in three reports [119,120,121]. Their ability to increase plasma LDL-C is a common feature acknowledged in different studies; in contrast to SFA, TFA intake may increase both LDL-C and HDL-C, unfavorably influence the LDL-C/HDL-C ratio, and hence raise the risk for CVD [72]. Moreover, increased dietary TFA frequently raises plasma triglyceride levels, lipoprotein (a), C-reactive protein (CRP), and proinflammatory cytokines, all of which are parameters pointing to a worsening of cardiovascular health in both healthy and diabetic subjects [122,123,124]. High TFA intake in humans is also linked to increased endothelial activation, such as increased levels of soluble intercellular adhesion molecule 1, soluble vascular-cell adhesion molecule 1, and E-selectin, possibly promoting vascular dysfunction [121]. Interestingly, controversial results have been obtained from clinical studies analyzing the impact of partially hydrogenated and ruminant TFA sources on CVD: although some researchers reported no differences, others have pointed to a more detrimental impact of partially hydrogenated TFA compared to ruminant sources, especially in terms of the lipoprotein profile [125].

### 2.2. Sterols

Sterols are a subgroup of steroids characterized by a common four-ring core structure, with a double bond between C5–C6, a saturated or unsaturated aliphatic chain in C17, and a hydroxyl group bound to C3. Based on their dietary sources, sterols can be divided into zoosterols (of animal origin) and phytosterols (obtained from plants), which chemically differ by an additional methyl or ethyl group on C24 or a double bond on the side chain, respectively [126]. The most relevant zoosterol is cholesterol, of which the circulating levels mainly depend on dietary intake and endogenous hepatic or extra-hepatic cholesterol synthesis [50]. The relationship between plasma TC and dietary cholesterol is currently controversial: observational cohort studies reported a linear association between diet intake and plasma TC levels [49], but it is worth noting that many confounding variables, such as the co-presence of SFA in many cholesterol-containing foods, could affect these observations. Several studies have been conducted to assess the impact of dietary cholesterol intake, mainly represented by egg consumption, on different CVD biomarkers. In this regard, Missimer and colleagues reported that increased dietary cholesterol raised both plasma HDL-C and LDL-C, without altering the LDL/HDL ratio, nor increasing the net CVD risk in 50 healthy young subjects [127]. Consistently, endothelial function-related parameters, including flow-mediated dilation, blood pressure, and body weight were not altered following different dietary intake amounts of cholesterol [128]. On the other side, interventional crossover studies have reported that egg consumption leads to higher plasma HDL-C levels and lowers various risk factors associated with CVD [129,130]. Hence, the current evidence regarding dietary cholesterol and CVD is still inconclusive.

Phytosterols can be classified as sterols and stanols, based on the presence of a double bond in the Δ5 position [131]. The effect of phytosterols on CVD-related parameters has been extensively investigated in recent trials. Observational studies show that subjects in the top quintiles of phytosterol intake have significantly lower TC and LDL-C levels compared to subjects with lower phytosterol consumption [132,133]. The reduction of intestinal cholesterol absorption due to competition for incorporation into micelles may at least partly explain the decreased LDL-C in plasma [134]. Interestingly, this positive effect upon lipid-related parameters was revealed after a few weeks of dietary consumption and seems to remain stable for a long time, although the long-term phytosterol-mediated effects on CVD risks are unclear.

### 2.3. Other Lipids

Vitamins A, D, E, and K are lipophilic, hydrophobic molecules assembled from isoprenoid structures, with essential biological functions.

Dietary vitamin A can be introduced either from vegetal foods in the form of carotenoids (e.g., β-carotene), or from animal-based foodstuffs, as esterified retinol (Table 1). Several studies have analyzed the association between circulating retinol and CVD [135], revealing a complex relationship with a characteristic U-shaped curve [136]. Accordingly, a recent cohort study upon hospitalized subjects for suspected CAD suggested that elevated retinol concentrations may be protective in terms of CVD risk. Conversely, extremely high retinol concentrations may affect CVD risk by itself or by negatively affecting other risk factors such as serum apoB and homocysteine levels [135]. Only a few studies have investigated the link between retinoic acid and CVD [53]: among these, 1530 patients with acute ischemic stroke in the upper 3rd quartile of circulating retinoid acid showed a reduction in CVD mortality as compared to patients with lower levels [137]. Further experimental evidence is still needed to clarify this potential positive modulation.

Although the main source of active vitamin D in humans is endogenous synthesis in the skin following sun exposure, it can also be obtained to a lesser extent from the diet (Table 1). Concerning the influence of vitamin D on cardiovascular health, observational studies have suggested that low circulating levels of 25-hydroxyvitamin D (25(OH)D) negatively impact on cardiovascular status [138]: in humans, vitamin D deficiency is associated with increased blood pressure, myocardial cell calcification, vascular dysfunction, and inflammation through various mechanisms, which are extensively described elsewhere [139]. However, recent intervention studies, meta-analyses, and randomized clinical trials have failed to demonstrate a beneficial effect of vitamin D supplements and CV outcomes in different populations [140,141,142]. In particular, the VITAL trial (vitamin D and omega-3 trial) assessed the impact of vitamin D3 (2000 IU/day) and ω-3 PUFA (1 g/day) supplementation in men and women aged ≥50 years, highlighting that vitamin D cannot be considered a protective factor against major CVD events [143].

The group of vitamin E is composed of α-, β-, γ-, and δ-tocopherols and tocotrienols (T3s). Different species of vitamin E are introduced completely through dietary sources (Table 1). Currently, extensive studies have been performed to investigate the role of vitamin E in CVD and prevention; recently, a long-term prospective cohort study on 29,092 subjects demonstrated that elevated serum concentrations of α-tocopherol are associated with a reduced CVD risk and overall mortality [144]. However, long-term supplementation of vitamin E and adverse cardiovascular outcomes has been extensively studied, yielding controversial results [145]. In contrast, preclinical studies suggest that in particular situations, such as acute myocardial infarction, which leads to an increased vitamin E requirement, vitamin E supplementation could help in preserving cardiac function [146,147].

## 3. Effects of SFA on RCT in Animal Models

There is a broad consensus that diets enriched in SFA increase the levels of HDL-C in humans [109,148]. However, the differences in the sources of fats used in these human studies do not always allow definitive conclusions to be drawn. The use of genetically homogeneous mice fed with well-controlled diets has confirmed the HDL cholesterol-raising effects of dietary SFA. Indeed, high SFA intake (13.4% *w/w*) induced a significant increase in HDL-C in C57BL/6 mice and in those expressing human cholesteryl ester transfer protein (CETP) and these effects were largely independent of the cholesterol content of the diet [28]. This change was also observed in C57BL/6 mice fed with an SFA-enriched diet (22% of caloric intake) compared with those fed with a micronutrient-matched low-fat diet [149]. An early study attempted to determine the mechanism underlying these HDL-raising effects of high SFA and cholesterol-containing dietary intake in both C57BL/6 and human apoA-I transgenic mice [150]. Indeed, turnover studies demonstrated that dietary SFA (11% of caloric intake) increased both the transport rate of HDL-cholesterol esters and apoA-I, whereas it reduced their fractional catabolic rate [150]. The authors observed the largest effect on the transport rate, but the specific effect of dietary cholesterol, which was included in the diet, was not addressed in that study [150]. In line with these findings, rabbits fed with a commercial chow diet plus 14% (*w/w*) coconut oil showed a two-fold increase in the HDL-apoA-I transport rate, but its catabolic rate was not changed [151]. Furthermore, the fractional catabolic rate of cholesterol from the HDL core appeared not to be affected by dietary SFA intake in C57BL/6 mice [28].

In this context, we also found that the expression of the major HDL-synthesis-related genes, such as *apoA-I*, *ABCA1*, and *LCAT*, was not affected by the intake of SFA in mice [28]. Interestingly, the plasma activities of the main HDL-remodeling enzymes, CETP and LCAT, were not affected by the ingestion of a diet that was highly enriched in SFA (16.5% *w/w*, mainly from coconut oil or butter) when compared with hamsters fed with another high fat diet (16.5% *w/w*), whereas PLTP activity was significantly higher in hamsters fed with coconut oil [152]. Consistently, a regular chow diet enriched with palmitic acid (15% *w/w*) did not affect hepatic LCAT expression and mass in mice expressing both human CETP and apoB100, although the main liver HDL receptor, SR-BI, was upregulated [153]. CETP activity was not affected by dietary SFA either (when changed from 5% to 20% *w/w*) in human CETP transgenic mice [154]. Since an SFA-enriched diet (13–18% *w/w*) had no significant effect on hepatic or intestinal *apoA-I* mRNA levels in both C57BL/6 and human apoA-I transgenic mice [28,150,155], apoA-I production appeared to be the main driver of SFA-mediated effects on HDL cholesterol levels in the different experimental animals reported, most likely at the posttranscriptional level [150].

Some researchers have attempted to determine whether variations in dietary fat produce changes in cellular cholesterol efflux to plasma and HDL. In an early study, African green monkeys were fed high fat diets containing either SFA (mainly from palm oil), MUFA, or PUFA, all providing 35% calories as fat, but the authors did not find any difference in the ability of HDL to induce cholesterol efflux from cultured fibroblasts [156]. The total efflux from macrophages to both plasma and HDL were moderately increased after feeding SFA-enriched diets to C57BL/6 mice [28,149], but in the first case it was largely dependent on the dietary cholesterol, and in the second case, the ABCA1-dependent efflux to HDL was reduced even after normalization by HDL cholesterol. More importantly, both studies independently evaluated the impact of SFA on the entire m-RCT pathway in vivo, i.e., the transport of radiolabeled cholesterol from macrophages to feces was traced in C57BL/6 mice and human CETP transgenic mice [28,149]. Hence, dietary SFA by itself did not accelerate the transport of macrophage-derived cholesterol to feces [28,149]; rather, on the contrary, reduced macrophage-derived cholesterol trafficking from the liver to the feces was observed following dietary SFA intake, concomitant with increased hepatic inflammation [149]. The main evidence of an effect of SFA on in vivo RCT is represented in Figure 2 and summarized in Table 2.

## 4. Effects of MUFA on RCT in Animal Models

In contrast with dietary SFA, MUFA appeared to improve many parameters of RCT, particularly when they were compared with those induced by SFA. Hence, a diet highly enriched with 15% trioleate (*w/w*) increased HDL-C and apoA-I levels, LCAT activity, and stimulated the production of the hepatic SR-BI receptor protein in rats when compared with animals fed with the same amount of fat derived from other sources [157]. However, the ingestion of a diet that was highly enriched in MUFA (mainly from canola oil, 10% *w/w*) did not affect LCAT, PLTP, or CETP activities in hamsters when compared with the animals fed with other high fat diets [152]. A regular chow diet enriched with oleic acid (providing 27% of the total calories from fat) also upregulated hepatic LCAT expression and mass, as well as SR-BI expression in human CETP and apoB100 transgenic mice [153]. Furthermore, dietary MUFA (12.6 % *w/w* of total fats) downregulated CETP expression and mass in the liver of human CETP transgenic mice [154]. Unfortunately, the effects of these changes on HDL-mediated RCT function were not addressed in these studies. Beyond its effects on HDL-remodeling enzymes, dietary MUFA (providing 12% calories) also enhanced ABCA1-independent efflux to HDL from C57BL/6 mice, although these changes were not observed when this parameter was normalized to the HDL cholesterol and even the ABCA1-dependent efflux component was reduced [149]. Importantly, dietary MUFA also promoted macrophage-to-feces RCT in vivo. These effects were mainly driven by the increased HDL levels and normal macrophage-derived-cholesterol trafficking from the liver to the fecal compartment compared with that of SFA-enriched diets [149]. It should be noted that we also found an enhanced macrophage-to-feces rate in C57BL/6 mice after administering 14 intragastric doses of a functional unrefined virgin olive oil enriched with its own phenolic compounds. This change was closely related with higher HDL cholesterol levels and an enhanced ABCA1-mediated cholesterol efflux to HDL [31]. However, the phenolic compounds present in the virgin olive oil appeared to be the main inducers of the macrophage-specific RCT rate, rather than the amount of oleic acid intake [31]. Furthermore, LDLR-deficient mice fed with a Western diet supplemented with 5% (*w/w*) of long-chain MUFA also showed accelerated ABCA1-mediated cholesterol efflux through the activation of *PPAR* transcriptional activity and concomitantly showed reduced atherosclerotic lesion sizes [158], thereby highlighting the beneficial effects of different MUFA on RCT. The main evidence of SFA impact on in vivo RCT is represented in Figure 2 and summarized in Table 2.

## 5. Effects of PUFA on RCT in Animal Models

The analysis of the mechanisms underlying PUFA-mediated cardioprotection has mainly been carried out in rodent models. Hence, PUFA (from 60% to 75% *w/w* of total fatty acids) strongly influences HDL metabolism in mice, rats, and hamsters [159,160,179] with a general effect of reducing circulating levels [30,159,160,179]. Moreover, the clearance of HDL is enhanced in different rodent models [159,179], with the latter mainly due to the upregulation of the hepatic HDL receptor, SR-BI. As for other sources of dietary fat, the impact of PUFA on the entire m-RCT pathway in vivo has been assessed in several independent studies. M-RCT was enhanced in mice fed with diets supplemented with either fish oil (menhaden oil, which is rich in ω-3 PUFA) [30] or simply after feeding with a supplementation of 1% ω-3 PUFA (*w/w* of total fats) in hamsters [161]. In line with these findings, biliary cholesterol secretion into feces was significantly increased in rats fed with 75% *w/w* of ω-3 PUFA of the total fats [160]. This enhancement was primarily due to increased hepatic-fecal excretion of HDL-derived cholesterol and it was consistent with an upregulation of hepatic ABCG5 and G8 transporters [30,161]. However, the favorable effect of PUFA (representing 32% of total fatty acids) on m-RCT was not confirmed in another independent study [180]. In contrast with the critical role of cholesterol on the entire RCT pathway [28], the effect of PUFA supplementation on m-RCT did not differ from that observed when saturated fatty acids were added to the high-fat, high-cholesterol diet [180]. Although this lack of effect contrasts with the stimulating effects of ω-3 PUFA C20:5 EPA and C22:6 DHA on the rate of fecal macrophage-derived cholesterol excretion [30,161], these results would be consistent with the absence of effects when comparing low- and high-soybean oil-enriched diets (which are rich in n-6 PUFA) in mice [30] or when evaluating the impact of alpha linoleic acid supplementation (representing 0.7% of calories from fats) in a high-fat diet in CETP-expressing apoE3 Leiden mice [165]. Importantly, this PUFA did not induce a further increase in the expression of ABCG5 and G8 in mice [30,180], rather suggesting differences in the bioactivity of PUFA species among different supplemented diets. Although not determined, the expected hepatic elevations in the cholesterol-derived oxysterols in mice fed with the high-fat, high-cholesterol diets in both studies [30,180] could also be hiding the potential favorable effects of ω-6 PUFA on *Abcg5* and *Abcg8* expression. In summary, the beneficial effects of ω-3 PUFA, rather than ω-6 PUFA, on RCT are well established. The main evidence of the impact of PUFA on in vivo m-RCT is represented in Figure 2 and summarized in Table 2.

## 6. Effects of TFA on RCT in Animal Models

Although many epidemiological studies are in agreement on the seriously adverse effects of TFA on human cardiovascular health, data from preclinical works are somewhat discrepant. The first evidence comes from a report by Gatto and colleagues, revealing that TFA may positively modulate lipid metabolism [162]. In fact, rats fed with a diet-enriched in various isomers of trans C18:1 (totally accounting for 32% *w/w* of total fatty acids) presented lower levels of total and LDL-C compared to animals fed with equal amounts of saturated or monounsaturated fatty acids. Despite not altering plasma HDL-C concentrations, a TFA-enriched diet caused changes in HDL composition (i.e., increased levels of TFA in the phospholipid fraction, at the expense of stearic and palmitic acids), without impairing the particles’ capacity to mediate RCT. The process was evaluated with a technique consisting of the intravenous injection of acetylated LDL radiolabeled in the cholesteryl ester portion; after a rapid distribution to the tissues, radioactivity re-appeared, reaching a steady state at 12–18 h. This study suggested that, in the absence of CETP, TFA may modulate the metabolism of apoB-containing lipoproteins without impairing the atheroprotective process of RCT [162]. Consistently, an independent study confirmed that a low amount of TFA (3% of total daily energy intake), as typically consumed in Mediterranean countries, does not affect RCT in mice, as demonstrated by the maintenance of HDL efflux capacity, LCAT activity, and cholesteryl ester hepatic uptake [163]. Even more surprisingly, the chronic consumption of high amounts (4.2% of total daily energy intake) of TFA (mostly C18:1) in the diet did not impact the expression of proteins involved in m-RCT, such as ABCA1, SR-BI, and HL in rats [164].

The impact of rumenic acid on m-RCT is more controversial. In the study mentioned above on humanized apoE3Leiden-CETP-expressing mice fed with a high-fat diet supplemented with CLA (representing 0.7% of calories from fats), cholesterol mobilization from macrophages to the plasma, liver, and feces was not improved compared with control animals [165]. Conversely, the presence of 4% *w/w* of total fatty acids of rumenic acid in milk administered to hamsters caused an increase in HDL levels and the upregulation of ABCA1 expression [166]. Although it was not directly assessed, both observations are expected to be related with the promotion of RCT in this experimental model. The main evidence of the impact of TFA on in vivo m-RCT is represented in Figure 2 and summarized in Table 2.

## 7. Effects of Sterols on RCT in Animal Models

Cholesterol and other sterols play an undisputed role in the homeostasis of cellular and systemic lipid metabolism, particularly within the cardiovascular system. In the last two decades, several studies have tried to address the impact of dietary sterols, such as cholesterol and phytosterols, on lipoprotein metabolism and more particularly on m-RCT. Unfortunately, because of the high heterogeneity of animal models used, experimental methodologies, and diet composition and duration, it is difficult to reach well-defined conclusions. Overall, studies have shown both an increase [28,29,168,169] and a decrease [167,170] in the m-RCT process or RCT molecular players following high-cholesterol dietary (from 0.2% to 2% *w/w*) exposure. On the other hand, the enrichment of diets with phytosterols (0.3–2% *w/w*) has generated more consistent results, highlighting effects such as reduced intestinal absorption and increased fecal elimination of cholesterol [171,172,181,182].

Mice were the most widely used animal model for both dietary cholesterol and phytosterols, although hamsters were also used, due to their natural expression of the CETP enzyme, which is lacking in mice.

As for cholesterol, an old study showed that C57BL6 were able to eliminate far less cholesterol from a muscle depot when fed with a diet rich in cholesterol (1.25% *w/w*), fat (15%), and sodium cholate (0.5%) compared to a chow diet [167]. The difference was also more pronounced when the C57BL6 strain was compared to the atherosclerosis-resistant C3H strain. Although this study does not quantify the specific macrophage-to-feces RCT rate, it highlights the importance of the mouse strain when designing in vivo RCT experiments.

A seminal study contributed to clarifying the crucial role of dietary cholesterol in upregulating the ABCG5 and ABCG8 transporters in the liver, thus promoting the disposal of sterols with feces [28]. In particular, the enrichment of a high-fat diet with cholesterol (0.2% *w/w*) significantly stimulated macrophage-to-feces RCT, measured according to the method standardized by Zhang, Zanotti, and colleagues [26], based on the macrophage-derived ^3^H-cholesterol percentage found in the mouse plasma, liver, and feces collected at fixed time points after the intraperitoneal injection of radiolabeled, cholesterol loaded macrophages. The enhanced RCT observed in high-fat/high-cholesterol-fed mice was independent of the saturated fatty acid dietary content, as well as of mouse sex, and was not associated with the development of obesity or insulin resistance. Notably, the same results were elicited also in transgenic mice expressing human CETP. The authors concluded that the hepatic upregulation of *Abcg5* and *Abcg8* transporters was the main mechanism leading to the increased RCT, since all the effects were completely blunted in ABCG5/G8-deficient mice. In this context, it is noteworthy that the increase of m-RCT might be a compensatory mechanism counteracting the atherosclerotic process, which is active in experimental models of diet-induced hypercholesterolemia.

Other evidence supporting the previous mechanism as a driver of the cholesterol-enhanced m-RCT comes from a recent paper [168], showing the upregulation of hepatic and intestinal ABCG5/G8 in C57BL/6 mice fed with a lithogenic diet (1.25% *w/w* cholesterol, 0.5% sodium cholate, 16% fat, 2% corn oil), compared to mice fed with a standard rodent diet. Similarly, feeding Wistar rats a 2% cholesterol-enriched diet resulted in increased intestinal expression of ABCG8, as well as liver X receptor α (LXRα), small heterodimer partner (SHP), and sterol regulatory element-binding protein 1c (SREBP-1c), compared to animals receiving the standard diet [169]. Though this study did not analyze the fecal excretion of sterols or bile acids, the increased expression of CYP7A1 found in the liver of cholesterol-fed animals suggests that there may be an increase in that process.

Important observations come from studies on hamsters. As stated above, this species naturally expresses CETP, although significant functional differences exist between the human and hamster enzymes, with the latter showing greater efficacy in transferring TG to HDL [183]. A first study assessed the in vivo m-RCT in cholesterol-fed (0.3% *w/w*) hamsters in comparison to standard chow-fed animals [170]. The dietary cholesterol enrichment caused a significant reduction of the ^3^H-sterols in feces, as well as in plasma and bile. The hepatic expression of key target genes was also modified, with *Abca1*, *Abcg1*, and *Abcg5* shown to be increased, whereas *Scarb1* and *Ldlr* were shown to be decreased. Furthermore, in vitro cholesterol efflux from macrophages to the plasma of cholesterol-fed hamsters was impaired [170]. Although these results suggest that diet-induced dyslipidemia may impair in vivo RCT in hamsters, a second study provided conflicting evidence. Briand F. and colleagues measured in vivo m-RCT in hamsters fed with a high-fat diet containing 0.5% cholesterol, 27% fat, 0.25% deoxycholate, and supplemented with 10% fructose in drinking water, in comparison with animals fed with a non-purified control diet [29]. In this experimental setting, cholesterol-fed hamsters showed higher ^3^H-sterol recoveries in their plasma, liver, and feces compared to controls. Conversely, ^3^H-tracer recovery in bile was lower, suggesting an impairment of hepatic cholesterol flux caused by the high-fat diet. The authors also found increased intestinal absorption of cholesterol in high-fat-fed hamsters, thus suggesting that the higher ^3^H-tracer recovered in feces may be due to a stimulation of the transintestinal cholesterol excretion. Finally, many RCT-related target genes, such as hepatic *Abcg5/g8*, *Sterol O-Acyltransferase 2 (Soat2)*, *Ld**lr*, and intestinal *Niemann-Pick C1-Like 1* (*Npc1l1)*, were downregulated in the high-fat diet group. However, it must be emphasized that the diametrically opposite effects on m-RCT shown in the two studies may be primarily due to the different composition of the respective diets. The higher fat content, as well as the integration of deoxycholate and fructose in the diet under study [29], can profoundly disturb the lipid metabolism in this animal model, in a synergistic or additive way to the effect exerted by cholesterol alone.

Contrary to the effects shown for cholesterol, no studies have evaluated the effects of dietary phytosterols specifically on the in vivo m-RCT. Most of the evidence is limited to the analysis of target gene expression, the quantification of sterols and lipids in key metabolic compartments, and the evaluation of their intestinal absorption. However, studies show interestingly consistent results, despite the heterogeneity sometimes found in their experimental designs. Mouse experiments showed that plant sterols and their 5α-saturated derivatives, stanols, are rapidly absorbed in the gut [182]. These compounds are then returned into the intestinal lumen, directly or via biliary excretion, thus competing with cholesterol absorption. Consumption of a phytosterol-enriched Western-type diet (2% *w/w* phytosterols, mainly β-sitosterol, and approximately equal amounts of campesterol and stigmasterol) for four weeks inhibited intestinal cholesterol absorption in transgenic atherosclerosis-prone mouse models, namely apoE- and LDLR-null mice, as well as in wild-type controls [171]. In this study, biliary levels of cholesterol were also reduced, whereas biliary bile acids remained unaffected. Dietary phytosterols reduced plasma cholesterol levels in atherosclerosis-prone mice only, suggesting that wild-type mice may compensate for their intrinsically low plasma cholesterol with increased synthesis. Importantly, the study showed that the observed effects were independent of the expression of core RCT genes, such as *ABC-transporters* or *NPC1L1*, which were only minimally changed. In a successive study, the same authors further examined the involvement of the ABCA1 transporter in the reduced intestinal cholesterol absorption observed in mice fed with the 2% phytosterol-enriched Western-type diet (PE-WD) [181]. The diet enriched in phytosterols did not modify the plasma levels of cholesterol or other lipids, nor did it alter the intestinal expression of genes such as *Abcg5/g8* or *Npc1l1*, regardless of the mouse genotype. Importantly, both wild-type and ABCA1-deficient mice fed with PE-WD for two weeks exhibited a significant decrease in intestinal cholesterol absorption compared with mice fed with control-WD, proving that this effect does not depend on ABCA1, at least in mice.

In line with these data, another study reported no changes in the expression of LXR target genes in both the liver and intestine of C57BL/6J male and female mice fed with a diet enriched in stigmasterol (0.3% *w*/*w*) compared with mice fed with a standard rodent diet [172]. However, the diet treatment promoted transintestinal cholesterol secretion, without affecting plasma or biliary cholesterol levels.

Overall, the current evidence suggests that further insights are needed in order to evaluate the specific effect of dietary sterols on reverse cholesterol transport in vivo. Most of the studies conducted do date did not allow us to clearly distinguish the relative contributions of cholesterol and phytosterols compared to other components of the diet. The dietary content of these lipids varies widely across studies and, more importantly, only a few (or none, as in the case of phytosterols) have specifically evaluated m-RCT. The main evidence of the impact of sterols on in vivo m-RCT is represented in Figure 2 and summarized in Table 2.

## 8. Effects of Other Lipids on RCT in Animal Models

Although the impact of liposoluble vitamins and their precursors and metabolites on atherosclerotic cardiovascular disease has been well established in humans [55,184,185] and animal models [185], their role in in vivo m-RCT is less characterized.

Dietary supplementation with either low (52 mg/kg) or high (129 mg/kg) doses of vitamin A affected several players in RCT in obese rats [173]. Both hepatic SR-BI and ABCA1 proteins were increased in the liver, a dual effect resulting in a decrease of circulating HDL. Moreover, vitamin A increased HL activity, possibly facilitating cholesteryl ester uptake by the liver via SR-BI. Notably, that study does not reveal a dose dependency, since both doses similarly affected the outcomes. This reduction of LCAT activity, together with increased SR-BI, may explain the observed reduction of HDL in animals fed with the standard diet. It is relevant to underline that this effect does not have negative implications, since the abnormal increased levels of HDL in obese vs. lean rats is related to dysfunctional particles. Notably, the evidence from that study raised questions about the actual impact of vitamin A on RCT, since the reported effects may imply dual consequences. Although the increase in hepatic SR-BI, despite reducing circulating HDL, resulted in an improved process [10], the overexpression of ABCA1 in the liver may favor the efflux of cholesterol back to the circulation, thus diverting it from biliary excretion [186].

The effect of astaxanthin, a carotenoid compound employed in dietary supplements for its anti-oxidant properties, on m-RCT was assessed through the classical method following radioactive cholesterol transfer from macrophages to the plasma, liver and feces. The supplementation of the diet with 0.5% astaxanthin for 2 weeks promoted m-RCT in wild-type and apoE-deficient mice [174].

In a seminal work, vitamin E, in the form of alpha-tocopherol, reduced the hepatic expression of SR-BI [175], with possible negative implications on the whole RCT process, which have not been investigated further. In contrast, two independent studies revealed a potential beneficial effect in atherosclerosis [176,177]. The administration of alpha-tocopherol 50–100 mg/kg/d for 4–8 weeks in rabbits fed with a high-fat diet or atherosclerosis-prone apoE null mice caused a reduction in the atherosclerotic lesion area. A putative mechanism, beyond the well-known anti-oxidant properties, is the stimulation of PPARγ-LXRα-ABCA1 axis, potentially leading to improved cholesterol efflux from vascular cells. It would be of interest to integrate these preliminary observations with the direct assessment of cholesterol mobilization throughout the m-RCT process.

The evidence relating vitamin D and RCT in vivo is limited to a study on hypercholesterolemic swine, in which the authors reported that dietary supplementation with vitamin D 1000 and 3000 IU/day dose-dependently increased the expression of ABCA1 and ABCG1 in the aorta. This result was supported by the observation that the in vitro incubation of THP-1 macrophages with 1,25 hydroxy vitamin D, the bioactive form of vitamin D, promoted cholesterol efflux. Altogether, these data suggest that vitamin D may exert antiatherosclerotic effects through the promotion of the first step of RCT [178].

The main evidence of the impact of vitamins on in vivo RCT is represented in Figure 2 and summarized in Table 2.

## 9. Conclusions

The impact of dietary lipids on HDL metabolism and cardiovascular risk has been deeply investigated in human studies [187,188,189,190,191,192]. Depending on the specific class of lipids and the ingested amount, both beneficial and detrimental effects on lipidemia and cardiovascular outcomes have been reported [94,95,193]. However, recent studies have challenged the classical view associating total and saturated fatty acids with increased cardiovascular morbidity and mortality [193,194,195], raising some concerns about the validity of the currently available nutritional guidelines. It is important to emphasize that results collected in the field of nutrition are more frequently derived from observational studies than from rigorous randomized clinical trials. This implies that definitive conclusions on the role of dietary lipids are difficult to reach, given the intrinsic variability of the population and their overall diets, in which other nutrients may affect the selected outcomes.

In this context, the availability of reproducible, accurate in vivo models is critical in order to provide an increased understanding of the mechanisms by which nutrients affect human health. The use of animal models to evaluate the m-RCT presents practical advantages and is of great value in order to obtain unique insights into all steps of this process and to assess their influence on the overall transport. Although the extrapolation to humans may not be rigorous, given the intrinsic differences between species, many efforts have been implemented to ameliorate the translatability of these results to humans. First, the application of the PREPARE [196] and ARRIVE [197] guidelines is currently recommended to improve the quality and reproducibility of studies involving animals. Second, the selection of human-like animal models (swine, hamsters, CETP- and apoB100-expressing mice, or liver-humanized mice [198]) that better recapitulate the lipoprotein metabolism in humans, may provide predictive results on the impact of dietary manipulations on the lipid profile and RCT in humans. Third, the critical analysis of in vivo data is mandatory in order to bridge the major findings derived from basic research in rodents to human pathophysiology and to accelerate the progress of our knowledge on the role of dietary lipids and develop strategies to reduce the risk of future cardiovascular events through a diet-based approach.

## Figures and Tables

**Figure 1 nutrients-13-02643-f001:**
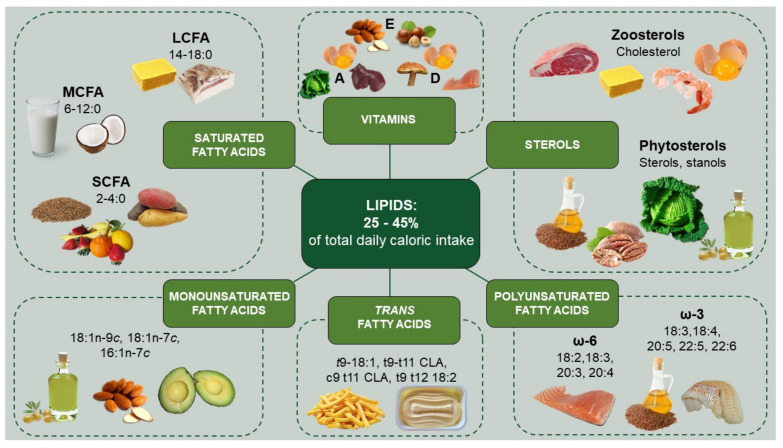
Main lipid classes and their dietary sources. Dietary lipids account for 25–45% of the total daily caloric intake and are classified into several subclasses based on their chemical structure. Saturated fatty acids are classified into SCFA, mainly obtained through the saccharolytic fermentation of undigested or partially digested fibers in proximal colon; MCFA, the main dietary source of which is represented by coconut oil and milk; and LCFA, found in large amounts in buttermilk, lard, and cocoa butter. MUFA are introduced into the diet through vegetable oils, mainly olive oil, sweet almond oil, avocado oil, canola oil, and others, whereas TFA are present in high amounts in partially hydrogenated vegetable oil and industrially processed food. The main dietary source of PUFA is represented by seeds and leaves of certain plants and cold-water fatty fish and fish oils. The main dietary source of cholesterol is represented by egg yolks, shrimp, meat, and buttermilk, whereas phytosterols such as sterols and stanols can be introduced through vegetable oils, grains and grain-derived products, and various nuts. Finally, dietary lipids can also be found in hydrophobic vitamins, such as vitamin A (bovine liver, egg yolk, spinach), vitamin D (egg yolk, cold water fatty fish and mushrooms), and vitamin E (hazelnut, olive oil, wheat germ oil). SCFA: short-chain fatty acids; MCFA: medium-chain fatty acids; LCFA: long-chain fatty acids; CLA: conjugated linoleic acid.

**Figure 2 nutrients-13-02643-f002:**
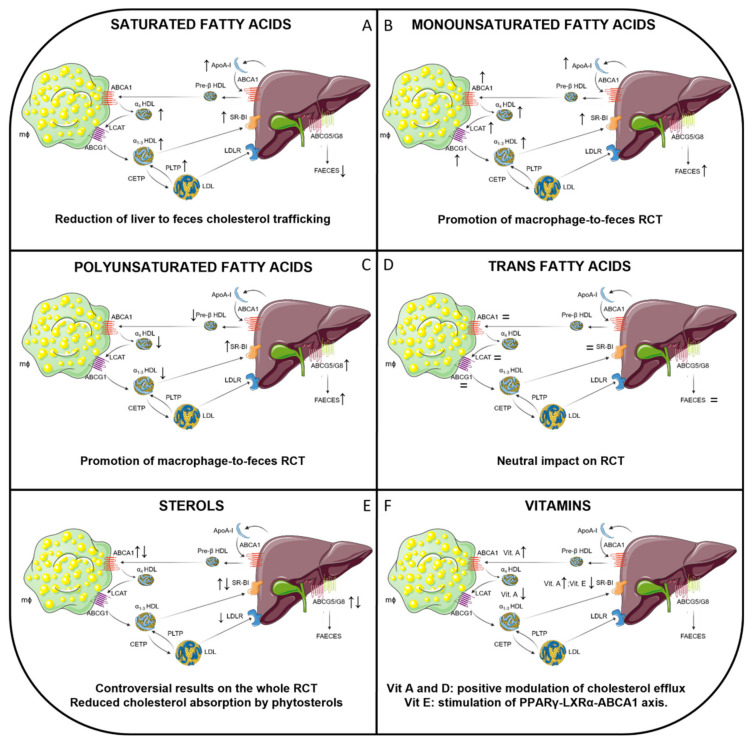
Impact of dietary lipid classes on RCT in animal studies. (**A**) SFA intake induces an increase in the HDL-C, apoA-I, and HDL transport rate in murine models, in parallel with a reduction in the fractional catabolic rate of HDL. Furthermore, a diet rich in palmitic acid increased PLTP activity in hamsters and SR-BI expression in mice expressing human CETP and apoB100. Overall, dietary SFA reduce cholesterol trafficking from the liver to the feces, negatively impacting on total RCT. (**B**) MUFA increase HDL-C and apoA-I levels, LCAT activity, and hepatic SR-BI expression in rats, whereas a diet enriched in MUFA did not affect LCAT, CETP, or PLTP expression in hamsters. Moreover, dietary MUFA enhance cholesterol efflux to HDL in C57BL/6 mice, as well as the macrophage-to-feces RCT rate. (**C**) PUFA intake induces a reduction in the serum HDL concentration in rodents, with a parallel increase in their clearance, probably due to an upregulation of hepatic SR-BI. PUFA, mainly ω-3 PUFA, also enhance macrophage-to-feces RCT in mice, with enhanced biliary secretion and hepatic ABCG5 and ABCG8 expression. (**D**) A low amount of TFA does not impact on RCT in mice, as demonstrated by unaltered HDL efflux capacity and LCAT activity; similarly, chronic TFA consumption does not affect ABCA1, SR-BI, or HL in rats. However, in hamsters, rumenic acid causes an increase in HDL levels and ABCA1 expression, possibly positively impacting RCT. (**E**) Sterols. Dietary cholesterol has been to observed both increasing and decreasing the m-RCT process or RCT molecular players such as hepatic ABCG5/G8, ABCA1, SR-BI, and LDLR. On the other hand, the enrichment of diets with phytosterols generated more consistent results, particularly by reducing intestinal absorption and increasing the fecal elimination of cholesterol. (**F**) Dietary supplementation with either low or high doses of vitamin A results in an increase in SR-BI and ABCA1 expression, resulting in decreased HDL in obese rats. Vitamin D supplementation promotes aorta ABCA1 and ABCG1 expression in hypercholesterolemic swine, suggesting a possible positive modulation of cholesterol efflux. Vitamin E reduces the hepatic expression of SR-BI; moreover, the positive modulation observed on the atherosclerotic lesion area in rabbits or apoE^-/-^ mice may be related to the stimulation of the PPARγ-LXRα-ABCA1 axis. apo: apolipoprotein; CETP: cholesteryl ester transfer protein; HDL: high density lipoprotein; LCAT: lecithin:cholesterol acyltransferase; LXR: liver X receptor; MUFA: monounsaturated fatty acids; PLTP: phospholipid transfer protein; PPAR: peroxisome proliferator-activated receptors; PUFA: polyunsaturated fatty acids; RCT: reverse cholesterol transport; SFA: saturated fatty acids; SR-BI: scavenger receptor class B type 1; TFA: trans fatty acids. Pictures were created by combining images from Smart Servier Medical Art (https://smart.servier.com, accessed on 14 June 2021). Servier Medical Art by Servier is licensed under a Creative Commons Attribution 3.0 Unported License (https://creativecommons.org/licenses/by/3.0/, accessed on 14 June 2021).

**Table 1 nutrients-13-02643-t001:** Dietary lipid sources, average daily intake, and main dietary sources.

Lipid Class	Daily Intake	Main Dietary Sources	Main Molecular Content	References
**SFA** SCFAs: acetic acid (2:0), propionic acid (3:0), butyric acid (4:0) MCFA: caproic acid (6:0), caprylic acid (8:0), lauric acid (10:0) LCSFA: myristic acid (14:0), palmitic acid (16:0), stearic acid (18:0)	7–10%	Refined coconut oil, Virgin coconut oil	Lauric acid (50%)	[33]
	7–10%	Buttermilk	Palmitic acid (28%), Stearic acid (12%), Myristic acid (10%)	[34]
	9 g	Palm oil	Palmitic acid (44%), Stearic acid (5%)	[35]
**MUFA** oleic acid (18:1 n-9c), vaccenic acid (18:1 n-7c), palmitoleic acid (16:1 n-7c), myristoleic acid (14:1 n-5c), erucic acid (22:1 n-9c)	15%	Canola Oil Olive oil Meat Peanut oil Sunflower oil	Oleic acid (57%) Oleic acid (68%) Oleic acid (30%) Oleic acid (50%) Oleic acid (45%)	[36,37,38,39,40]
**ω-3 PUFA** ALA (18:3), stearidonic acid (18:4), EPA (20:5), DPA (22:5), DHA (22:6)	1.5 g	Flaxseeds Chia seeds Canola oil Soybean oil	ALA (57%) ALA (20%) ALA (10%) ALA (5%)	[41,42]
	250–500 mg	Cod liver oil Salmon Tuna	DHA + EPA (8.3 g + 10.8 g *) DHA + EPA (1.19 g + 0.89 g *) DHA + EPA (2.15 g + 0.8 g *)	[43]
**ω-6 PUFA** LA (18:2), GLA (18:3), DHGLA (20:3), AA (20:4)	10 g	Soybean oil Sunflower oil Corn oil Nuts Peanut butter Seeds	LA (67.7 g *) LA (50 g *) LA (50 g *) LA (34 g *) LA (13,45 g *) LA (4 g *)	[44,45]
**TFA** Elaidic acid (t9 18:1), trans vaccenic acid (t11 18:1), CLA, rumenic acid (c9 t11 CLA), t10 c12 CLA, linoelaidic acid (t9 t12 CLA)	1%	Industrial bakery Meat, dairy Meat, dairy	t9,t11-CLA c9,t11-CLA t10,c12-CLA	[46,47,48]
	1%	Industrial bakery	Elaidic acid t-18:1	
**Zoosterols** Cholesterol	300 mg	Egg yolk Shrimp Cod liver oil Buttermilk Bovine liver Lard	1337 mg * 150 mg * 570 mg * 250 mg * 194 mg * 95 mg *	[49,50,51]
**Phytosterols** Campesterol, sitosterol, campestanol, sitostanol, stigmasterol, stigmastanol, brassicasterol	150–400 mg	Canola oil Olive Chickpeas Sunflower oil	Campesterol (156 mg *) Sitosterol (158 mg *) Sitosterol (42.3 mg *) Sitosterol (171 mg *)	[49,52]
**Liposoluble Vitamins**	700–900 µg	Bovine liver Buttermilk Egg yolk Carrots Yellow pumpkin	Vit. A (16 mg *) Vit. A (906 µg *) Vit. A (607 µg *) Vit. A (1.15 mg *) Vit. A (599 µg *)	[53]
	20 µg	Herring Tuna Sardines Salmon Egg yolk	Vit. D (30 µg *) Vit. D (16 µg *) Vit. D (11 µg *) Vit. D (8 µg *) Vit. D (5 µg *)	[54]
	15 mg	Wheat germ oil Sunflower seed oil Hazelnut EVO Oil	Vit. E (136 µg *) Vit. E (49 µg *) Vit. E (25 µg *) Vit. E (21 µg *)	[55]

Daily intake is expressed as a percentage (%) of the total daily calorie intake or total amount per day, and may slightly vary depending on age, sex, and condition (e.g., pregnancy, breastfeeding). The main molecular content in dietary sources is expressed as percentage (%) of the total food content or as the total amount per 100 g of the food product (*). AA: arachidonic acid; ALA: α-linoleic acid; CLA: conjugated linoleic acid; DHA: docosahexaenoic acid; DHGLA: dihomo-γ-linolenic acid; DPA: docosapentaenoic acid; EPA: eicosapentaenoic acid; EVO: extra virgin olive oil; GLA: γ-linolenic acid; LA: linoleic acid; LCFA: long-chain fatty acids; MCFA: medium-chain fatty acids; MUFA, monounsaturated fatty acids; PUFA, polyunsaturated fatty acids; SCFA: short-chain fatty acids; SFA: saturated fatty acids; TFA: trans fatty acids; Vit: vitamin.

**Table 2 nutrients-13-02643-t002:** Impact of dietary lipid classes on lipid profiles and RCT in animal studies.

Lipid Class	Animal Model	Dietary Treatment and Duration	Effect on Lipid Profile	Effect on RCT	Reference
**SFA**	Mice (C57BL/6 and human CETP transgenic; males and females)	LFLC diet with high SFA content (saturated fat/total fat ratio 0.64) for 8, 16, 24 weeks	↑ HDL-C; ↑ ApoA-I	↑ m-RCT	[28]
	Mice (C57BL/6J; males)	SFA-HFD (45% kCal from palm oil) vs. micronutrient-matched LFD for 24 weeks	↑ HDL-C	↑ cholesterol levels in liver and feces; ↑ total cholesterol efflux to plasma and HDL	[149]
	Mice (C57BL/6 and ApoA-I Tg; males and females)	HFD with 27% *w/w* SFA for 5 weeks	↑ HDL-C; ↑ ApoA-I	Not evaluated	[150]
	Rabbits (New Zealand White; females)	Chow diet + 15% *w/w* hydrogenated coconut oil for 3 months	↑ HDL-C	↑ HDL-ApoA-I transport rate	[151]
	Hamsters (Golden Syrian; males)	Chow diet supplemented with 16.5% *w/w* SFA (coconut oil and butter) for 6 weeks	↑ HDL-C; ↑ ApoA-I; ↑ PLTP activity	Not evaluated	[153]
	Mice (Double Tg expressing human CETP and apoB100; males)	Chow diet enriched with 15% *w/w* palmitic acid for 4 weeks	= TC; = LDL-C	↑ Hepatic SR-BI expression	[153]
**MUFA**	Rats (Sprague-Dawley; males)	Chow diet enriched with 15% *w/w* trioleate for 20 days	↑ HDL-C; ↑ ApoA-I	↑ LCAT mRNA and activity ↑ SR-BI hepatic expression	[157]
	Hamsters (Golden Syrian; males)	Chow diet supplemented with 10% *w/w* MUFA (canola oil) for 6 weeks	↑ TC; ↑ non HDL-C; ↑ HDL-C	Not evaluated	[154]
	Mice (double Tg expressing both human CETP and apoB100; males)	Chow diet enriched with oleic acid for 4 weeks	↓ TC; ↑ HDL-C; ↑ LDL-C	↑ LCAT mRNA; ↑ SR-BI hepatic expression	[153]
	Mice (CETP-Tg; Sex n.s.)	low-fat (5%) or high-fat (20%) diets containing olive oil (enriched in MUFA) for 2 weeks	= TC; ↑ HDL-C	↓ CETP expression and mass	[154]
	Mice (C57BL/6J; males)	MUFA-HFD (45% kcal from sunflower oil) for 24 weeks	↑ TC; ↑ HDL-C	↑ m-RCT; ↑ ABCA1-independent cholesterol efflux to HDL from C57BL/6J mice	[149]
	Mice (LDLr−/−; females)	Western diet supplemented with 5% *w/w* LC- MUFA for 12 weeks	= TC; = HDL-C; = LDL-C; = TG	↑ ABCA1-mediated cholesterol efflux to ApoB-depleted plasma	[158]
**PUFA**	Mice (C57BL/6; females)	Regular diet enriched in ω -3 FA for 16 weeks	↓ TC; ↓ TG; ↓ HDL-C;	↑ Hepatic SR-BI expression; ↑ Hepatic uptake of HDL-CE	[159]
	Mice (C57BL/6J; females)	Diet supplemented with either low SO, high SO, CO, or FO for 6 weeks	Not reported	↑ m-RCT (FO diet compared to high and low SO diet ↑ Hepatic ABCG5/ABCG8 expression induced by FO diet	[30]
	Rats (Wistar; males)	Diet supplemented with sunflower oil (ω–6) or fish oil (ω –3); duration n.s.	↓ TC; ↓ TG; ↓ HDL-C	↑ Biliary cholesterol secretion = biliary phospholipids = bile salts	[160]
	Hamsters (Golden Syrian; males)	HFD enriched in ω-3 for 20 weeks	↓ TC; ↓ TG ↓ HDL-C;	↑ m-RCT ↑Hepatic ABCA1, ABCG1, SR-BI, ABCG5/ABCG8 expression ↓LCAT activity	[161]
**TFA**	Rats (Sprague-Dawley, males)	Various isomers of C18:1 TFA versus equal amounts of SFA or MUFA for 4 weeks.	↓ TC; ↓ LDL-C; = HDL-C ↑ of TFA in HDL phospholipids	↓ Hepatic [^3^H]-cholesterol	[162]
	Mice (C57BL/6J, males)	Low amount of TFA (3% total daily energy intake as trans 18:1 fatty acid) for 7 weeks	= TC = HDL-C ↑ TG	= Cholesterol efflux to plasma from mice = LCAT activity; = Transfer of CE to liver by HDL	[163]
	Rats (Fischer, females)	High amount of TFA (4.2% total daily energy intake) versus MUFA/PUFA-containing diets for 52 weeks	↑ of TFA in plasma phospholipids	= Hepatic SR-BI, LDLr, ABCA1 expression	[164]
	Mice (apoE3Leiden-hCETP, males)	HFD supplemented ± CLA or ALA for 12 weeks	= plasma lipids	= m-RCT	[165]
	Hamsters (Golden Syrian, males)	Milk fat diets ± rumenic acid	↑ HDL-C ↓ TG	↑ Aortic ABCA1 expression	[166]
**STEROLS**	Mice (C57BL/6J and C3H, males and females)	HFD + 1.25% cholesterol + 0.5% cholate for 4 weeks	↑ TC ↓ HDL-C	↓ RCT (fecal elimination of cholesterol from a muscle depot)	[167]
	Mice (C57BL/6J, males and females; other transgenic strains were also used in this study)	HFHC diet versus different control diets for 8 weeks	↑ TC ↑ HDL-C ↑ non-HDL-C =TG	↑ m-RCT	[28]
	Mice (C57BL/6J, males)	Lithogenic diet (1.25% cholesterol, 0.5% sodium cholate, 16% fat, 2% corn oil) for 8 weeks	Not evaluated	↑ RCT ↑Hepatic and intestinal ABCG5/G8 expression	[168]
	Rats (Wistar, males)	Cholesterol rich diet (2% *w/w*) for 2 weeks	↑ TC ↑ VLDL-C ↑ LDL-C ↑ HDL-C	↑ Intestinal ABCG8, LXRα, SHP, SREBP-1c ↑ Hepatic CYP7A1	[169]
	Hamsters (Golden Syrian, males)	Cholesterol enriched diet (0.3% *w/w*) for 4 weeks	↑ TC ↑ HDL-C ↑ non-HDL-C ↑ TG	↓ m-RCT ↓ cholesterol efflux capacity of chol-fed animal plasma ↑ Hepatic *Abca1, Abcg1, Abcg5* ↓ Hepatic *Scarb-1, Ldlr*	[170]
	Hamsters (Golden Syrian, males)	HFD + 0.5% cholesterol + 0.25% deoxycholate + 10% fructose in drinking water for 4 weeks	↑ TC ↑ HDL-C ↑ non-HDL-C ↑ TG	↑ m-RCT * * controversial, because of the impaired hepatic cholesterol flux ↓ Hepatic *Abcg1, Abcg5, Ldlr, Acat2* expression	[29]
	Mice (ApoE -/-, females	Western type diet ± 0.5, 1% or 2% phytosterols (mainly β-sitosterol, and equal amounts of campesterol and stigmasterol) for 4 weeks	↓ TC ↓ VLDL-C ↓ IDL-C ↓ LDL-C (in the 2% phytosterol group)	↓ biliary cholesterol	[171]
	Mice (C57BL/6J, males and females)	Standard diet ± 0.3% stigmasterol for 4 days	=TC	↑ transintestinal cholesterol secretion	[172]
**Vitamin A**	Rats (leand and obese WNIN/ob; males)	Diet supplemented with low (52mg/kg) or high (129 mg/kg) doses of Vitamin A for 20 weeks	↓ TC; ↓ HDL-C	↑LXRα, RXRα hepatic expression ↑ABCA1, SR-BI, HL hepatic expression only in obese rats	[173]
	Mice (C57BL/6 and ApoE -/-; males)	AIN-93G diet supplemented with astaxanthin (0.05%, *w/w*) for 2 weeks	↑ HDL-C ↓ non-HDL-C	↑ m-RCT	[174]
**Vitamin E**	Rats (Wistar; males)	Chow diet depleted of α-tocopherol for 28–40 days, followed by 400 mg/kg refeeding of vitamin E for 48 h	= TC; = HDL-C	↓ Hepatic SR-BI expression	[175]
	Mice (ApoE -/-; males)	Chow diet supplemented with vitamin E; 4–8 weeks	= TC; = TG	↓ Aortic CD36 expression ↑ Aortic PPARγ, LXRα, ABCA1 expression	[176]
	Rabbits (albino; males)	Vitamin E-poor diet, vitamin E-poor diet with 2% cholesterol, or vitamin E-poor diet containing 2% cholesterol with daily intramuscular injections of vitamin E (50 mg/kg) for 4 weeks	↑ TC in rabbits fed with diet supplemented with 2% cholesterol compared to controls	↑ PPARγ, ABCA1 expression in rabbits that underwent intamuscolar injection of Vitamin E	[177]
**Vitamin D**	Hypercholesterolemic miniswine	HCD supplemented with 1000 IU/day or 3000 IU/day Vitamin D vs. controls for 48 weeks	Not reported	↑Aortic ABCA1 and ABCG1 expression	[178]

↑ increase; ↓ decrease; = no change; ABCA1: ATP Binding cassette subfamily A member 1; ABCG: ATP Binding cassette subfamily G; ALA: α-linoleic acid; Apo: apolipoprotein; CD36: cluster of differentiation 36; CE: cholesteryl esters; CETP: cholesteryl ester transfer protein; CLA: conjugated linoleic acid CO: corn oil; FA: fatty acid; FO: fish oil; HCD: high cholesterol diet; HDL: high-density lipoprotein; HL: hepatic lipase; HFD: high fat diet; IDL: intermediate-density lipoproteins; LDL: low-density lipoproteins; LCAT: lecithin:cholesterol acyltransferase; LDLr: low-density lipoprotein receptor; LFLC: low-fat, low-cholesterol; LXR: liver X receptor; MUFA: monounsaturated fatty acid; n.s.: not specified; PLTP: phospholipid transfer protein; PPAR: peroxisome proliferator-activated receptors; PUFA: polyunsaturated fatty acid; m-RCT: macrophage-to-feces reverse cholesterol transport; RXR: retinoid X receptor; SFA: saturated fatty acid; SO: soybean oil; SR-BI: scavenger receptor class B type I; TGs; triglycerides; TC: total cholesterol; TFA: trans fatty acids; Tg: transgenic.

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
