# Peer review of "Impact of Dietary Lipids on the Reverse Cholesterol Transport: What We Learned from Animal Studies"

_nutrients, 2021, doi:10.3390/nu13082643_

Round 1

Reviewer 1 Report

My comments of this paper was as follows: I don't have any special comments, but if you could mention the upper limit level of dietary lipids in terms of inverse transport of cholesterol or a guideline for intake, it would be helpful in the prevention of cardiovascular disease.

Author Response

Thanks you for your comment. Indeed, the use and outcome of animal models is pivotal to bridge the translational gap to the clinic by providing valuable data on the expected therapeutic performance of a certain therapy. However, although animal models will help to build the required safety data package of dietary supplements, we believe that it is hard to suggest dietary guidelines from preclinical studies. The variability in the diet composition and experimental outcomes, as well as the paucity of studies where dose-responses were investigated, make the translation very difficult.

On the other hand, the correlation of the dietary intake of nutrients with specific outcomes is very important to provide insights into the mechanisms relating diet with lipidemia and RCT. Therefore, to better inform the readers on the amount of fats employed in each work, we mention this information in the text, where not originally present.

Reviewer 2 Report

This is an interesting review on what is currently known about a) the mechanisms of action of the various classes of dietary lipids along with b) a review of how animal models have provided insights into how the reverse cholesterol transporter functions and how various dietary lipids may impact on the RCT.

There are several areas where the manuscript falls down for which I will make point by point comments-

1) For example the sub heading for Section 2- General aspects of dietary lipids, chemistry sources, intake and effect on CVD. It would be useful to add in that this is in humans. The same applies for all of the other sub headings throughout the manuscript-if it can be clear which sections focus on knowledge obtained from clinical studies in humans versus those which focus on the animal models

2)  There seems to be some significant limitations with many of the animal models when it comes to elucidating the impact of dietary manipulations on lipid profiles as well as the RCT activity. It would be useful if this information could be tabulated (and then alluded to in the text), to make it easier for the reader to understand the limitations.

3) There is not enough focus on what directions the research in animal models needs to now take, including what needs to be done to ensure that the data obtained are more likely to be able to be translated into humans. More emphasis needs to be given on the complex interplay of all of the dietary lipids which makes it difficult then to untangle the precise impact of manipulating the diet for one particular type of lipid on the RCT in an animal model. 

4) No mention is made of exercise which is another factor which may impact on RCT. Are there any animal models where both dietary manipulation along with exercise undertaken to see if there was an impact on the RCT?

Author Response

Point 1. For example the sub heading for Section 2- General aspects of dietary lipids, chemistry sources, intake and effect on CVD. It would be useful to add in that this is in humans. The same applies for all of the other sub headings throughout the manuscript-if it can be clear which sections focus on knowledge obtained from clinical studies in humans versus those which focus on the animal models

Response 1. According to the reviewer’s suggestion, now we mentioned in the corresponding section’s titles whether the reported data came from either clinical (human) or experimental studies.

Point 2.  There seems to be some significant limitations with many of the animal models when it comes to elucidating the impact of dietary manipulations on lipid profiles as well as the RCT activity. It would be useful if this information could be tabulated (and then alluded to in the text), to make it easier for the reader to understand the limitations.

Response 2. Following the reviewer’s advice. Data on figure is also shown in a new table (Table 2) whereby the effect of dietary lipids on lipid profile and RCT has been summarized. Additionally, data on table 2 is also properly discussed in the corresponding section of the revised version of the manuscript.

Point 3. There is not enough focus on what directions the research in animal models needs to now take, including what needs to be done to ensure that the data obtained are more likely to be able to be translated into humans. More emphasis needs to be given on the complex interplay of all of the dietary lipids which makes it difficult then to untangle the precise impact of manipulating the diet for one particular type of lipid on the RCT in an animal model.

Response 3. The criticism raised by the reviewer on the difficulty to isolate the effect of individual lipids is correct and it applies either to studies in animals and humans. We commented these limitations and discussed potential improvements of the studies involving animals in the paragraph of conclusions.

Point 4. No mention is made of exercise which is another factor which may impact on RCT. Are there any animal models where both dietary manipulation along with exercise undertaken to see if there was an impact on the RCT?

Response 4. We are aware that many additional factors, both from diet and lifestyle, may impact RCT efficiency. However, in this review, as well explained in the title, we decided to unravel the role of dietary fats in influencing this atheroprotective process. Physical exercise, as well as other dietary components or nutraceutical interventions are beyond the scope of the present manuscript.

Reviewer 3 Report

The authors summarized that the effects of dietary lipids such as FAs, sterols, and other lipids on the RCT in animal models. The reviewer agrees to the authors’ opinion that the use of animal models to evaluate the m-RCT is of great value, so the significance of this review is great.

Main comments

Figure 2 is main Figure in this review, but it is difficult to understand the contents. For example, it is better that showing increase/decrease in targets rather than showing the targets using yellow square. In addition, it is unclear whether Figure 2 shows the result of human-like models (hamsters and CETP-expressing mice) or mice. How about summarizing the results for each animal in Table? Figure 2 needs to be reconsidered.

Author Response

Point 1. Figure 2 is main Figure in this review, but it is difficult to understand the contents. For example, it is better that showing increase/decrease in targets rather than showing the targets using yellow square. In addition, it is unclear whether Figure 2 shows the result of human-like models (hamsters and CETP-expressing mice) or mice. How about summarizing the results for each animal in Table? Figure 2 needs to be reconsidered.

Response 1. We kindly agree with reviewer 3 in that compelling data on figure 2 in a table may contribute to better understand the impact of each dietary approach on circulating lipids and lipoproteins and RCT. Accordingly, we modified the figure 2 and included a new table (Table 2), showing in detail the animal model, dietary approach, effects on blood lipid and/or lipoproteins, and RCT.